# Bibliodrama: Introducing Stories from Narrative Traditions in the Development of Young People's Life Orientation

**Jean Agten** †

Religious and Secular Worldview Education, Odisee University of Applied Sciences, Warmoesberg 26,
1000 Brussels, Belgium; jeanagten@telenet.be
† Retired Lecturer.

**Abstract:** Young people, in the age of puberty and early adolescence, are in need of images and narratives as role models to mirror their actual thoughts and feelings, and to stimulate the development of their (tradition(s)-related) life orientation. The development of a life orientation we see as a religiously or secularly founded process of identity construction—a work-in-progress; a process of 'learning by doing'. This is described in Part I of this contribution. As Jacob Moreno, the founder of psychodrama stated: 'Thinking is in the action'. 'Doing'—being actively involved in a situation—is the defining characteristic of key persons and key objects in narratives. This is in line with John Dewey's view that the activity new perspectives are created; learning by doing. In bibliodrama, making use of psychodrama techniques, connections are established between narratives from traditional (religious or secular) worldviews—but also from myths and fairy tales—and young people's individual life experiences. Seemingly without effort and as child's play, bibliodrama creates an encounter between the here-and-now and the once-and-then of narratives of long ago. The theoretical framework of this practice and the methods of bibliodrama are described in Part II. In Part III we present an example of bibliodrama performed with 18–20-year-old students of the Odisee Hogeschool (Odisee University of Applied Sciences) of Brussels (Belgium). We conclude our contribution with a few preliminary conclusions, a discussion and recommendations for the practice of bibliodrama in education to familiarise students with (religious and secular) life orientations traditions, in order to facilitate the construction of their own authentic life orientation.

**Keywords:** identity construction; life orientation; bibliodrama; narratives; interreligious encounters; education

---

## 1. Life Orientation as Part of Identity Construction

### 1.1. Communicating with the Multivoiced Self

The development of a life orientation is 'an existential positioning process pertaining to the meaning of the human being, the world and the meta-empirical, directed towards the horizon of the good life' [1] (p. 45). Life orientation as an identity construction can be approached in different ways. In this contribution we will go deeper into the concept of 'identity' as a dynamic process. In our approach, identity is seen as dialogically constructed in a multivoiced self [2,3]. Listening to the variety of voices in the multivoiced self requires awareness of, and attentiveness to, these inner voices—voices that represent warnings, invitations or evoke feelings of wonder. Children and adolescents gain insight in their many inner voices as pieces in a colourful mosaic of (religious and secular) worldviews or life orientations—as these are represented in the multivoiced self [3]. By telling about memories that are stored in their minds, children structure these memories and make preliminary connections between

the various fragmentary images, feelings and thoughts: a dynamic process of identity construction. On the one hand, the telling and retelling of experiences changes the narration itself; on the other hand, it also transforms the related feelings, thus contributing to the process of (religious) identity construction. Identity-under-construction is like a never-ending story. Through the process of retelling, the narrator constantly finds him/herself in a new position in the restructured narration: a flexible temporary balance. The narrator is not only the 'protagonist' or the main character of the story, she/he can also more or less be seen as the author of her/his own story. However, as we will see below, each person's narrative is also constructed by the way she/he is assigned a role/position in the narrative of 'the other', as is each story contextually and historically situated.

In addition to the 'inner' conversation, by Hannah Arendt [4] coined as 'thinking', as an important aspect of identity construction, this construction process includes—according to Vygotsky [5]—important 'outer' (social) characteristics. From that point of view, children and adolescents are co-constructors of their own identity. The construction of a young person's 'puzzle' becomes clearly visible in an activated (un)conscious comparison with others. This comparison process consists of listening, exploring, questioning, accepting and respecting, copying, recognising, rephrasing, distinguishing, conflicting and hyphenating—aspects elaborated upon by Marcia [6] in his description of the stages of exploration and commitment in identity development. The expressions, actions and attitudes of others offer many opportunities for recognition and/or acceptance/endorsement as valuable input for the individual process of identity construction. Exploring the input of others and changing perspectives stimulates the individual processes of identity construction (as work-in-progress). At the same time, young people redesign and adapt the input of others to meet their own developmental needs. As we will show below, the process of identity construction is facilitated by narratives, which create the possibility to catch one's breath in the safe space of bibliodrama. At successive moments in time in a session, the bibliodrama process contributes to the construction of a uniquely biographically shaped identity, which remains continuously susceptible to new experiences and the stories of others. Narrativity structures life events and gives meaning to a person's identity-under-construction (see also Verhofstadt-Denève [7]).

## 1.2. Communicating with Religious Narratives and Practices

The interactive process of co-constructing (that is, in the presence of the other) a (religious or secular) worldview or life orientation should not be a mere topping of one image and perspectives of individuals with another image or perspective. This process needs to be enriched by the way worldviews and life orientations are practiced by relatives and friends in accordance with how they are represented in a person's wider context. Religious narratives from existing and living traditions may evoke memories of, and reveal connections with, a person's own life experiences. These narratives can provide elements for the ongoing process of identity construction, i.e., elements that are incorporated in an individual's personal narrative—in a reworked form or not. In addition, such elements can further be integrated in the ongoing narrative of the respective tradition as well. The activity of remembering contributes to the rewriting of one's own biography, but also contributes to new insights in the tradition. A person's own story, which includes words, images and fragments of a particular tradition, becomes part of that tradition, while the story simultaneously reconstructs the worldview/life orientation of that person. Within the framework of a rich variety of (religious and secular) worldviews, not only traditional narratives should be included, but also—and especially— examples of actual practices (experiences, actions and attitudes). In addition, symbols of the beginning of a religious awareness and partial manifestations of current (re)new(ed) life orientation(s) play an important role in a person's religious identity construction.

## 1.3. Tradition—An Ongoing Narrative

The word 'tradition' refers to the Latin *tradere*, which means 'handing over', 'passing on' and 'supplying with information'. Children and (young) adults do not adopt entire (religious or secular) worldviews that are passed on to them in their process of identity construction. Making use of religious

traditions, they selectively make use of passages and construct their own images based on their own interpretation of these (passages from) traditions, and use them as building blocks for their identity. Young people are touched by, and create their own interpretations of, the variety of worldview passages they encounter in the plural context(s) in which they live. These images become meaningful due to the feelings they evoke and the ways they are verbalised in conversation with others [8]. In dialogical encounters, the religious literacy of young people is increased and their life orientation is articulated, while at the same time they improve their social positioning in their peer group, which is important for the (further) development of their own positionality amidst the plurality of life orientations. Just like young people, traditions (religious and secular) are constantly challenged to reformulate their being-in-the-world, responding to the multivoicedness of constantly renewing personal narratives.

## 2. Bibliodrama—Theory and Method

### 2.1. Definition of Bibliodrama

The Concept of Bibliodrama

The word 'bibliodrama' refers to the Greek biblio, which means 'book', in our case, more specifically, books in which religious and secular traditions are recorded, such as the holy scriptures of Christianity and Islam, and myths and fairy tales. Drama literally means 'action' or 'what happens'. This brings to mind theatre and other forms of expression. In bibliodrama, it is (part of) a biblical, Qur'anic or mythical narrative that is put into action as a theatre play. However, the vision and methods used in bibliodrama are not limited in their application to religious narratives. Narratives from different cultures, or even stories lifted from the real-life experiences of the participants, can form the starting point for a bibliodrama. As such, a bibliodrama session can contribute in a comprehensive manner to intercultural and interreligious communication. The type of bibliodrama that is presented in this contribution aims at open communication and the sharing of any kind of stories that are deemed important and valuable, and which are part of the makeup of the participants' biographies. Bibliodrama as a method for the exploration of narratives will be illustrated in this article with a religious narrative taken from the Sufi tradition. (In the original use of the word 'bibliodrama' (by Zacharias), the word 'biblio' refers to the 'Bible'. I use the word here in a broader sense. In the European context, the word bibliodrama is usually employed only in connection with Bible stories. In my view we should not limit the word bibliodrama to the Bible, but apply it in a broad way as referring to 'books'.)

Many scholars used the metaphor of drama when writing about life, Shakespeare being one of them:

All the world's a stage

And all the men and women merely players;

They have their exits

and their entrances

And one man in his time

plays many parts

His act being seven ages

(Quote from: Shakespeare, *As You Like It* (II.vii).)

The definitions of bibliodrama range from 'man who plays many parts', the role-playing of main characters of stories—as mentioned in the quote of Shakespeare—to the re-reading of selected scenes where each participant reads the lines of a character out loud. In this contribution we follow the description that the psychotherapist Peter Felix Kellermann has given of psychodrama, which we adapt for bibliodrama [9] (p. 20). In the context of this article, bibliodrama is seen as

a method in which participants are invited to continue and complete their actions through dramatisation, role-playing and dramatic self-presentation. Both verbal and nonverbal communication are used. A number of scenes are performed, depicting for example specific interpretations of traditional (religious or secular) narratives. The scenes may approximate real-life situations or can be externalisations of inner processes. Usually, three phases can be identified in a bibliodrama session: warming-up, action and sharing.

The Dialogical Self Theory (DST) [3] provides a consistent framework for bibliodrama. DST does justice to the multivoiced character of each person's identity, and the mixture of feelings that can accompany a person's positionality vis-à-vis real-life situations, persons and objects. In DST, a person is seen as an unsurpassable story teller, who by her/his narrating capacity constructs her/his biography and identity [2].

### 2.2. The Method of Bibliodrama

Bibliodrama is performed in a restricted space, a 'stage' set up in the middle of the room where the participants gather. Before the drama can start, in a prephase, the participants are introduced into the concept of bibliodrama, the possible interaction of a character's qualities and interpretations with one's own (in)competencies and experiences, the right not to play a part in staged scenes and the restrictions regarding sharing the experiences outside the safe space of this particular group of participants. The first phase of the enacted drama is the so-called 'warming up'. A 'warm up' may include some physical exercises, a round of 'how are you today' or—if the bibliodrama session is part of a series of sessions—a review of the previous meeting. In the example given below, the bibliodrama session starts with offering a story to the participants, followed by an invitation to open up to the narrative and letting themselves be touched by it, and to reflect on why it has that effect and what the possible relation is with their own life story.

The second phase of the role-playing and performance of the narrative starts at the moment when one of the participants identifies with one of the 'roles' in the story, and positions her/himself on stage (see also Sundèn [10]). The role-playing ends at the moment when the participants distance themselves from the character they have played.

The final phase of bibliodrama is the sharing phase, during which the participants exchange experiences and tell each other about the personal meanings they assign to the performed (fragment of the) narrative.

At the end of the day, a bibliodrama continues after the actual bibliodrama session took place, by establishing relationships between and gaining (better) insight in the staged story and situations in the participants' daily lives.

In the phase of role-playing and acting, participants on stage relate to narratives and characters in a structured and playful way. The rules of the game, in this case the structures of bibliodrama, guarantee the intimacy/confidentiality of a safe, constructed space, and, accordingly, provide each participant with the possibilities of 'doing as if . . . '. The leader of the play, the so-called 'director', monitors compliance with the rules, and, by extension, watches over the safety of the created space; a safe space that offers the possibilities for exploration and creativity. Within this safe space elements of the real lives of the participants come to the fore, tensions can be experienced, laughter can be part of the play, frustration may arise, or feelings of relief may be experienced. 'Safe spaces are 'contentious' and 'risky', yet 'playful' and 'pleasurable' and ripe with educational possibilities . . . ' (Stengel & Weems, in [11] (p. 58)).

Once the playing stage is over, the participants put aside the role of the character they played and reflect upon the play from a personal perspective in the sharing phase. The (re)presentation of the experiences during the play are not discussed, nor the lessons learned for a participant's daily life. The impressions that are shared during the third phase are never addressed again during informal conversations 'outside'. Literally: neither outside the safe space of the bibliodrama nor outside of the room where the play was performed are these impressions ever revisited. Nevertheless,

there is of course an effect on the daily activities of the participants. A lot can be learned from an executed bibliodrama, which is precisely why people are eager to be included in a bibliodrama: to literally practice another perspective, to reflect—from the historical perspective of a biblical/Qur'anic character's point of view—on existential dilemma's that are age-old, and that emerge just as well in the contemporary lives of the participants. In this sense, bibliodrama can be regarded as a laboratory for imagination, experimentation and exploration that creates awareness of concerted actions and resistance, narratives and counter-narratives.

Bibliodrama is not about the literal role-playing of a narrative from the Bible or the Qur'an for example. Nor is it about performing a well-designed script, like in a theatre play. Bibliodrama is a form of natural improvisation, a spontaneous exchange between the narrative and the participants. The participants give shape to their character or the situation by including elements from the selected parts of the narrative, while simultaneously—unconsciously and intuitively—including their own biographically related interpretation. It is precisely in this way that the encounter between the individual life stories of the participants and the narrative is facilitated. In bibliodrama, both body and mind are activated on stage, but also real-life experiences and memories, dreams and longings, doubts and strict convictions.

Bibliodrama is a creative method for participants who are looking for spiritual experiences and meaning in life. The focus is on shared meaning construction within the encounter between personal experiences and narratives (for example from the Bible, the Qur'an or national and international literature). The method aims at exploring existential issues in a creative and playful way by incorporating contextual information from the participants' own lives and the societal context in which they exist [12]. Through use of imagination and by having a lived, theatrical experience of situations, bibliodrama brings narratives into existence and turns inspiration into a bodily experience. Imagination, drama, role-playing, puppet play and a variety of bodily arts are all 'research instruments' in bibliodrama. The participants role-play a character from a selected narrative and 'stage' this character according to the way this character's position has become/is becoming a 'voice' in their 'society of mind' [3,7].

Bibliodrama connects today's real-world experiences with age-old stories. This is why sharing the experiences—at the end of the play, during the sharing session—is given full attention. The participants are assisted in recognising the—until now probably unnoticed—connections between the narrative, their role-play and own daily experiences. Recognising own experiences in the shared remarks can lead to a feeling of relief, or even result in a confirmation of the 'rightness' of their emotions for the participants. New lines of thought may develop out of the sharing stage, new relations that were previously invisible to the participants they may now notice. Mutual support may emerge from the recognition of not being the only one to have these kinds of tense experiences. The sharing phase is the moment to link the play experiences with experiences in daily life, and the societal reality in which the participants live. Every bibliodrama session needs to wind up in a sharing session.

### 2.3. Bibliodrama: A Safe Space—Freedom within Boundaries

Although in bibliodrama improvisation is the core element of role-playing, the structure is clear and a professional 'director' is needed. Bibliodrama is characterised by an open and inductive approach to narratives, and by a group-oriented approach that facilitates interaction and the collaborative expansion of a repertoire of thoughts, feelings and actions. Preconditional is an open attitude towards intergroup and inter-worldview conversations. Each participant is given the space to play/perform a character or situation, and has the right to interpret a character in her/his own unique way.

The playing space has clear boundaries—literally and metaphorically—safeguarded by the 'director', the facilitator. The participants play/represent a character, an object or specific scene from the narrative, which is demarcated from their actual 'I', while at the same time interpreted on the basis of their real 'I' and actual biography. The facilitator's task is to support the 'actors' in their emotionally charged experience of the play, to guarantee that they are able to withstand the fullness of the

performance. If necessary, the facilitator must be able to protect the 'actor' against her/himself, making sure that they can—at any time—remain the master of their own expressions, emotions and actions.

Not only the 'actors', but also the text of the narrative itself are safeguarded during the role-play. The 'director' preferably is an expert in hermeneutics, that is, she/he is familiar with different 'schools' of interpretation of texts—be their origin in religious, philosophical or other narrative traditions. The 'director' monitors the bibliodrama session and at the same time offers possible interpretations and favourable action patterns that are implicitly included in the story. The facilitator presents these patterns as mirrors, as mere possibilities for playing/acting. There is never any persuasion or coercion. In all circumstances, the players/actors are given the freedom not to follow the comments, remarks, instructions or whatever else the facilitator provides. The 'director' never disqualifies the comments of the participants but follows the interactions and counteractions with an open mind.

In bibliodrama, each participant has her/his own pace. While for some people a breakthrough occurs on the spot—during the performance, for others clarifying insights emerge in the days following the bibliodrama session. In the next section, we will look more closely at this aspect of 'lifelong learning', which is induced by bibliodrama.

### 2.4. Stories in Bibliodrama

In this section we elaborate on the function of 'stories'. In bibliodrama an encounter is arranged between the story located in an historical space, as imagined in virtue of the narration, and the person as 'actor', located in an actual space and reality. These spaces come together and unite, bringing imagination and reality, the past and the present together. For the interaction of re-imagining and remembering and reality, the actual situation, the child psychiatrist and therapist Winnicott coined the concept of 'intermediary space' [13]. While playing with a doll, the child reproduces and relives a particular situation, making this event re-occur in line with its imagination, through which the child (re-)finds its positionality in the respective situation. Playing with the doll enables the child—consciously or unconsciously—to get (more) grip on the situation in question and become the storyteller of its own story. This is the core of what happens in bibliodrama.

Below we focus on human beings as motivated storytellers [2] (p. 1), and the role of stories of others in the construction of people's own narratives, autobiographies and identities.

### 2.5. Characteristics of Bibliodrama

#### 2.5.1. Living and Learning through Emotions

Bibliodrama offers a language for experienced feelings. While playing and sharing experiences, emotions, memories and images emerge. The language that is about and takes place through emotions—emotionally charged language [14]—needs to be taught, just like reading and writing and arithmetic. Emotional intelligence, as researched by Goleman [15], is the personality dynamic or the potential that needs to be nurtured and developed in a person; "emotional literacy is the constellation of understandings, skills and strategies that a person can develop and nurture from infancy throughout his or her entire lifetime" [16] (p. 11). In our view, illiteracy with regard to articulated expressions of emotions can be seen as a problem of our time [16] (p. 9). Facilitators in bibliodrama have an important task in offering verbal formulations and concepts for identifying emotional experiences during play. A vocabulary is needed to answer questions like 'How do you feel right now, in this scene of the play?', or 'What kind of emotion do you feel right now?' and 'Tell me, where in your body do you notice the physical reaction to the situation, and how would you qualify it?' In an emphatic way, the facilitator can offer words to enable a participant to express verbally what is at stake for her/him at a given moment, words which are just as applicable to moments in real-life that may coincide with the situation enacted during the bibliodrama session.

Physical changes in a person's appearance, pointed out by the facilitator, increase the awareness of the meaning of the situation—of the situation during play, as well as similar situations occurring

in real-life. For example, observations like 'You're looking away', 'You're fiddling with your skirt' or 'I see your hands are shaking' increase the awareness of the meaning of emotions at stake in the situation for that person. By making explicit what would otherwise remain implicit, the possibility of exploration arises, which results in an increased level of awareness during day-to-day concerns.

In our view, becoming emotional literate regarding careful and nuanced expressions of emotions is an important aspect of education for children and young people in their development—in particular the development of life orientation—during puberty and early adolescence. Bibliodrama contributes to the development of emotional literacy, and subsequently to the articulation of a life orientation.

### 2.5.2. Living and Learning through Action

Performing, playing and acting out (different fragments of) a narrative breathes life into the story's characters. They are given a voice by the participants—now 'actors' on the stage. The 'actors' behave as if they are the characters themselves. 'As if'—the childlike competence to play and pretend—is crucial in bibliodrama. 'As if' denotes the participant's personal truth about the character from the narrative. While moving their bodies, the 'actors' are—in a figurative sense—moved to explore the words and actions of the character, and of their co-actors. The 'actors' learn to listen, to see other people's needs, to feel free to touch each other, to ask for help and to offer assistance. In bibliodrama they experiment with looking after others and taking care of them—a kind of behaviour that can be practiced in the daily lives of all the participants.

### 2.5.3. Living and Learning through Topical Matters

Bibliodrama can be seen as a method to explore the meaning of texts from religious and secular narrative traditions for individuals living in a secularising/secularised age. The meaning of a story emerges in a person's (listener's) encounter with a narration (respectively read or told out loud). The re-told narrative is performed by the 'actors' in their own unique way, colouring the narrative with their own language, attitudes and actions. The tradition comes to life, acquires a flavour of actuality and becomes real for every 'actor' involved. The past becomes a present for the player(s), indicating ways to deal with future situations.

### 2.5.4. Living and Learning in Life Orientation

By playing their role in ancient stories from various religious and secular traditions, the 'actors' experience and gain insight ('action insight') into what is really at stake in these narratives. Simultaneously, they may also become aware of the similarities and differences with their own life. Role-playing means recognising, exploring, purifying and practicing, creating new images and possibly expanding one's repertoire of interactions, which can serve as an orientation in daily life, a life orientation.

The narratives, being told and retold in bibliodrama, reveal tracks for giving ultimate meaning to life, and salvific paths as these are experienced from generation to generation. The 'actors' follow these tracks, position themselves in the narrative tradition, and discover new orientations and new horizons for themselves.

### 2.6. Types of Bibliodrama

Many methods, many styles of performing, and many aims can be thematised in a bibliodrama session. Some bibliodramas are person-oriented, while others are group-oriented. Bibliodrama can revolve around religious narratives or personal life stories. The focus can be on personal religious development, or the development of a religious community aimed at open and interreligious relationships with the neighbourhood, for example. Some bibliodramas come close to psychodrama and are therapeutic in their methods. Bibliodrama can be staged in a theatre like in 'Playback theatre' [17]. It can also be used to facilitate individuals' exploration of religious scriptures, and is drawn upon in many forms of pastoral care, catechesis, religious education, Bildung, art and meditation—not only for

children, but for adolescents, adults and senior citizens as well. Bibliodrama as a theoretical concept covers many practical methods.

### 2.7. Encounter(s) in Bibliodrama

By way of its inductive and open communicative character, bibliodrama (in its broadest sense) creates a space of encounter for individuals with all kinds of religious and secular backgrounds and their personal search for meaning. The way in which this is done is described in the following sections.

### 2.7.1. Linking Up with the Participant's Life World

The starting point for the bibliodrama session is the person's initial reaction to the story, even if this is a negative reaction, such as 'what a dull story' or 'this story does not appeal to me at all'. During the first encounter with the story—after a collective reading—attention is paid to these initial reactions: questions asked, feelings of anger, sadness or happiness that arise, remarks that are made or emerging doubts. These first responses indicate how each participant can participate in a common theme, and how this theme might contribute to the development of the personal life orientations of the participants. For the facilitator, the initial reactions of the participants are also an indication of the level of support and coaching that will be needed for this specific group. The facilitator's own biography in turn plays a role in this process of signalling/diagnosis, coaching and offering support.

### 2.7.2. Exploring the Richness of the Narrative

A story is constructed out of a variety of perspectives and actions of characters, vividly coming to mind with the help of objects, images and experiences—any of these might touch upon the memories of the participants. Bibliodrama makes room for a diversity of possible meaning constructions, and offers possibilities to identify with a variety of aspects included in the narrative. Of course, not all possibilities can be explored in one bibliodrama session. During each session, however, priority is given to the elements of the narrative that the participants are willing to explore in collaboration.

### 2.7.3. Exploring the Interpretation(s) of the Participants

In bibliodrama the 'actor' is allowed, and even stimulated, to shape the characters and situations from the narrative according to her/his own unique interpretation of the character. In this way a connection is established between the 'actor's' self-understanding and the character. In this process, she/he is assisted by the other participants. The confrontation between the participants' interpretation of a character and an 'actor's' staging of this character stimulates a deeper understanding and strengthening of the participant's own positionality in life. It opens up the possibility of rewriting the script of one's life. In this way the participants develop their self-understanding and positionality in the group, in their network, and ultimately in the larger society.

### 2.7.4. From Chronos to Kairos

In bibliodrama, time changes from clock time to experienced time, from chronos to kairos. The 'director' of the bibliodrama slows down the actions of the participants, for example, by repeating out loud what they say, by asking them to physically adopt the posture they think fits their line or by asking another participant to do so. In the latter case, an 'actor' is invited to ascribe words to what she/he sees and what kind of emotions she/he feels by watching the 'scene'. Decelerating the performance in this way facilitates the awareness of the 'actor' and stimulates interaction and communication with the other participants. By proceeding in this manner, more than just one interpretation and meaning of the story comes to the fore, enlarging in this way the repertoire of points of view and the flexibility of all the participants.

### 2.7.5. Acting out a Story—Stories in Action

In bibliodrama, a story is retold and relived in the 'staging' and the interactions between the participants. Participants' memories of earlier experiences with people or objects shape the staging in a playful way, and conversely, the staging enriches their perspective on earlier experiences and their memories. Each participant has to make 'acting' choices, has to position her/himself vis-à-vis the particular articulation of the theme that is the subject of the session. These moments exert influence on their life orientation and inspire and motivate their positionality in real-life—sometimes in an impressively radical and far-reaching way.

### 2.7.6. The Communicative Frame of Reference

A (religious or secular) life orientation perspective sheds its light on everything that moves the 'actor' in bibliodrama to choose a 'role' and to identify with that 'role'. The personal experience becomes part of a larger whole, which includes the other 'actors' and their counterpositions in guise of the opponents in the play. The way in which the other participants give meaning to their counter-role further shapes the 'actor's' own role interpretation and stimulates further self-understanding. The personal worldview identity is communicated—within the framework of the staged narrative—alongside others point(s) of view, giving way to the reorientation and reconstruction of their personal positionality in terms of life orientation.

### 2.7.7. Giving Way to Existing Frames of Reference

It is obviously possible in bibliodrama that the interpretations of a narrative by scholars and by participants do not align, in the same way as the interpretations of scholars who conduct exegesis do not run parallel and even change over time, possible even in the course of a lifetime. By creating a space for young people/students to freely represent a narrative, the possibility is created to explore traces of interpretations and to go off the beaten track. New ways of interpreting may arise immediately as the 'show' is being put on stage. It is also possible that at a later point in life the intensity of the staging experience suddenly emerges in a surprising way, at an unexpected place and time.

### 2.7.8. Interreligious Communication

In bibliodrama narratives from a variety of worldviews/life orientations pertaining to the participants' cultural or religious backgrounds become recognisable and understandable, leading to more respect for each other's background. Similar stories from different life orientation traditions can be explored—not for the mere activity of exploration but to contribute to the mutual enrichment of the thinking and acting repertoire. The different perspectives that emerge—due to the diversity of the participants and their life orientations—likewise offer many new angles, both for the participants and for the ongoing narrative tradition(s).

The method of bibliodrama, in particular, creates space to deepen one's positionality in a familiar religious or secular worldview tradition—be it Judaism, Christianity, Hinduism, Buddhism or Islam—through interreligious encounters. In that sense, by mirroring, the development of one's own worldview related identity is articulated and the commitment thereof strengthened.

## 3. Bibliodrama in Action—Interreligious Identity Construction

This part of our article focuses on a concrete example of a bibliodrama, performed with 18–20-year-old students of the Odisee Hogeschool (Odisee University of Applied Sciences) of Brussels (Belgium). Special attention is paid to aspects of the inter-religious communicative skills of young people that are stimulated by narration, improvisation and 'staging'. We conclude this part with recommendations for the further development of bibliodrama practices in education.

In the session described below, the narrative of 'River and Sand' (This specific story is part of the literary treasure of the Sufi tradition, a spiritual 'school' within North African Islam.) takes centre-stage,

with its thematic content related to aspects that come to the fore in the identity development processes of young people. In this narrative, through a changing context, different positions are presented, which can be seen as a metaphor for identity exploration in a plural society [6]. It is in particular the exploration of modes of communication that is set on stage, envisioned as a stimulus for rewriting the script of personal life orientation.

*3.1. River and Sand*

Once upon a time there was River. She had her source high up in the mountains. Playful and lean she swung herself through the valleys. On her way down she became increasingly stronger and wider. Then she reached Desert. Upon arriving at Desert she wanted to flow right through Sand. However, River noticed that all her water disappeared, no matter how quickly she tried to flow through Sand. She was convinced that it was her destiny to cross Desert, but she didn't perceive a way to reach her goal.

Now the hidden voice of Desert whispered to her: 'Wind can cross the Desert, and so can you!' River however had never before heard Desert's voice. 'I'm afraid I'll get mixed with Sand and be absorbed by Sand,' River replied. 'Wind can fly but I cannot.' 'As you are now, you won't be able to cross it,' whispered Desert. 'You would disappear or turn into a swamp. You must allow Wind to carry you across Desert, to your destination.'

'Wind can't do that,' River scoffed. 'Oh, yes,' said Desert, 'You must allow Wind to pick you up and carry you.' River could not accept such a crazy idea. A river high up in the air! She would lose her personality, and who could guarantee that she would ever become herself again?

Desert said, 'I promise you will be you again. Wind will pick you up and carry you across Desert, and then she'll let you go. You'll fall down like raindrops, and then your water will turn into a river again.'

'How can I be sure you're not lying?', River asked. 'It's the truth!', said Desert. 'And if you refuse to believe me, bad things await you.' 'Why can't I stay as I am now?', River protested. 'It can't be done!', said Desert, 'but who you really are will not be lost. On the other side of Desert you'll receive the same name, because what you are, remains!'

Then River surrendered to the welcoming arms of Wind. Tender and swiftly he carried her up. When they reached the peaks of the mountains on the other side of Desert, Wind gently dropped River. And all her drops flowed together, looked for each other, and became a stream again ... then a small river ...

Then River asked Desert: 'How did you know it would be like that?' And the Sand of the Desert whispered: 'We knew it, because we see it happen day after day, and because we, Sand, stretch all the way from Desert to the mountains. That's why people say that the way in which the flow of life continues its journey, is written in sand' (From Shah [18]).

*3.2. The Encounter of River and Sand*

As an example of bibliodrama in actu, we selected one fragment of the story of River and Sand to be turned into a performance for young people.

3.2.1. 'Staging the Narrative'

'Staging the narrative' is based on the scene from the narrative 'River and Sand' in which two 'characters' (River and Sand) meet each other. The participants, now becoming 'actors', give shape to this scene in verbal and nonverbal ways that fit their imagination and their own interpretation of the characters in the narrated situation. In the encounter that is put on stage, tension is visualised by the 'actors' as they physically move towards each other—a tension resulting from a mixture of emotions, such as curiosity about each other, longing for each other, fear of the otherness of the other, doubt as to whether one is welcomed by the other or turning away from the other. Such an encounter bears within it the possibility of conflict, serenity, disarming smiles, liberation, etc. First and foremost, the encounter may be the beginning of something new, a recovery, a renewed admission or the sowing of seeds of deepening emotions related to the theme that emerges in the play.

Two participants, taking on the role of 'actors', take up their part in the play. They are interviewed by the 'director' of the play. Questions like 'Who are you in this narrative?', 'Where do you come from?' or 'What do you feel at this moment, as the narrative starts?' Then the 'actors' enter the stage—the encounter begins. The way this encounter is shaped, flows from the personal interpretation and inspiration of the 'actors' and their mutual interaction—verbal and nonverbal. Both shape their character and play their 'role' on stage according to the way they are touched by the narrative, and in line with the image of the character they have created in their minds and hearts while listening to the story.

The encounter is then 'staged' a few times with different 'actors'. This allows for the emergence of different perspectives on this particular encounter, and, accordingly, for the emergence of different perspectives on encounters in everyday life—and encounters in general. The variation in the performances makes the participants aware of their unique way of taking on a role and giving shape to it, and also of the reactions of the other 'actor' to their role, i.e., of the attitude and behaviour of the opposite 'actor' on stage. In addition to stimulating awareness, the plays also serve as an instrument for exploring and experimenting with new insights and the behaviours that result from these. The plays do not take up much time, they are brief and to the point. They are kept short, especially for groups with new participants, to familiarise them with this particular way of 'staging a narrative'.

### 3.2.2. Playing with Motives

By 'acting', the participants experience what inspires and motivates them in their actions, what they really long for and what hinders them in realising their dreams. The 'director' of the play invites the participants to listen attentively—and with their heart—to the part of the narrative that will be put on stage, and to take the time to reflect on that aspect.

### 3.2.3. Identification

All of the participants—whether they are directly on stage or not—are invited to take on the role of the character that touches them the most, the character they feel most committed to and would prefer to play—in this example, the role of River or the role of Sand/Desert.

### 3.2.4. Re-Reading the Narrative

As a 'warming up', the part of the narrative that is central to the play is re-read, and the participants are invited to listen attentively—to let their hearts speak—and to activate their imagination to create the character once they are on stage.

Once upon a time there was River. She had her source high up in the mountains. Playful and lean she swung herself through the valleys. On her way down she became increasingly stronger and wider. Then she reached Desert. Upon arriving at Desert she wanted to flow right through Sand. However, River noticed that all her water disappeared, no matter how quickly she tried to flow through Sand. She was convinced that it was her destiny to cross Desert, but she did not perceive a way to reach her goal.

### 3.2.5. On the Stage

The first duo of actors prepares for the encounter between River and Sand. The others take up their position as 'audience'.

Sand positions himself in the middle of the stage, usually marked by a carpet. River positions herself in one corner of the stage, and slowly walks in the direction of Sand. The participants who take on the role of audience have an overview of the stage. Their task is to engage in introspection, i.e., to pay attention to their inner world and to explore what touches them in the encounter between River and Sand. In what way does this encounter touch upon their personal experiences and what kind of memories does this scene evoke about (inter)relationships in their daily lives?

The action should be brief and concise. If there is any conversation besides the nonverbal aspects of the communication, it may be reduced to just a few short sentences or statements, perhaps two or three expressions from both 'actors'.

3.2.6. Imagination at Work

With help of the 'director', both 'actors' are stimulated to familiarise themselves with their role. Taking turns, and prompted by the questions and comments of the 'director', they tell who they are, how they feel, where they come from and what their goal is.

When people participate in a bibliodrama for the first time, the 'director' will handle in-depth questions carefully and tactfully during the 'interview'. Little by little, during a series of bibliodrama sessions, the director may receive permission from the participants to ask more intrusive questions. The participants will get used to opening themselves up in a space of trust, and accordingly their competence in 'acting out their insight' will increase. Below, we present a number of questions that can facilitate participation in the staging of a character:

* Please, tell me, where do you come from? (aimed at presenting oneself and one's origins; Are you a native? Are you alone? Who are your comrades?);

* What kind of River are you—a tiny stream, a swirling River, ... ? (aimed at presenting one's identity);

* What is your position in the context? What does your 'natural' context look like? (aimed at clarifying one's positionality in the natural world and the societal context);

* What is your goal? (aimed at the verbalisation of one's life orientation; What or who would you like to be?).

3.2.7. The Beginning

The participant who represents River starts the play, and then moves to the middle of the stage, facing Sand. River expresses the feelings she goes through while moving. Next, Sand responds to the approaching of River's water, i.e., River's attitude and the verbal expressions of her feelings. Both 'actors' do their utmost best to stage an encounter and to accompany their action(s) with verbal expressions. They give way to feelings of joy, anger or whatever comes from the heart. After a few minutes, the 'director' signals the end of this staging of the encounter. Each play is concluded with a sharing session, during which experiences are exchanged.

3.2.8. Sharing Experiences

The 'actors' shake off their role (sometimes by literally shaking their heads, arms, hands and legs) and reposition themselves in the circle of participants/the audience. During this sharing stage, all participants—with the two 'actors' coming in last—reveal what they have experienced, what kind of emotions were evoked and what kind of memories from their personal lives were triggered by the play.

After the first 'staging' and 'sharing' phase, the cycle repeats itself: a re-reading of the narrative, followed by imagining, identifying, performing and sharing.

3.2.9. Examples of Playful Encounters

In this section we give examples of how the theory of bibliodrama can be translated into practice, and illustrate its method as set out above. After providing a brief introduction to the participants (students of the Odisee Hogeschool/Odisee University of Applied Sciences, Brussels, Belgium), a bibliodrama was set up with them based on the narrative of 'River and Sand'. Below we give some examples of the actors' verbal comments during the scene when River and Sand run into each other:

* River says: 'Why are you holding me back, Sand?' Sand answers: 'I'm not holding you back at all, you're running into me. You can start with saying 'hello!"

* River walks towards Sand and says: 'Hello Sand, you know your way around here, I'd like to stream through but I don't know if that's a good thing?' Sand says: 'The rivers I meet dry out or become swamps, I don't know what I trigger in them, but apparently my very nature doesn't bode well for rivers.'

* River: runs very fast and bumps into (the player in the role of) Sand. Sand says, 'Didn't you see me? I've been around here for quite a while, you know.'

Below we provide some examples of the encounter between River and Sand where crossing the boundaries of identity is at stake.

* River says: 'Wow so much sand, what is all this sand doing here, that doesn't belong here, this is my terrain'. Sand asks: 'And why don't I belong here?' River says: 'You make me sink, you have to respect me'. Sand answers: 'You make me wet with your water, so that I become mud or a swamp, I can't be expected to become a river, can I?'

* River: 'Damn, I can't move forward anymore, it feels like I'm drying up'. Sand says: 'You're losing yourself, you're not following your own riverbed anymore, are you?' River says: 'I'm constantly making new beds'. Sand says: 'But you're trying to flood me, you can't just do that, you have to watch out where you flow.'

* Sand calls out to River as she approaches: 'River, River, look out! Stop, you're going to sink, disappear'. River says, 'What do you mean, I'm going to disappear? I'll see that happen first, you don't know me yet!'. Sand says: 'That's right River, I don't know you, but I know me. The way you come pouring from the mountains, that's something worth seeing, I don't want you to disappear. We must deliberate.' 'Deliberate, deliberate! Move away you mean. You're the one that needs to move Sand, so that I can pass through. Ask Wind for help to blow you away.'

### 3.2.10. The Sharing Session—Exchange of Experiences and Feelings

In this concluding part of the bibliodrama, the question is: What kind of memories, associations, feelings emerged while playing your role and while you were an observing member of the audience? What struck you in particular? What touched you? What surprised you? What was recognisable? During this round of questions, no comments are allowed on the way the 'actors' performed the scene.

The focus in the sharing phase is exclusively on expressing one's own associations and feelings and sharing them with the other participants. First, a participant verbalises her/his feelings while she/he played the role of River or Sand. Then, the same participant expresses her/his personal feelings. For example: 'The moment Sand said . . . , for me, as River, a feeling of . . . . emerged.' Such a statement may then be followed by a personal comment: 'This feeling was very strange to me. In daily life I usually don't allow myself to have those kinds of feelings'. It is in no way allowed to say something like: 'When you (pointing to one of the participants) said to me . . . But that's not true.' Or: 'The way you were River and made such a blunt statement, that's exactly as I know you when you are among friends.' Judgmental expressions are not permitted—positive ones or negative ones.

The same applies to the members of the audience. They should also be very attentive and express only what they have experienced themselves. For example: 'The way you (pointing to one of the participants) played Sand reminded me of my own experience of blocking someone's way, which gave me a feeling of . . . '. A strict distinction must be kept in mind between the staged role and the person of the 'actor'.

Below we elaborate on the sharing session. We present a few more statements, all of which show how dramatically the narrative triggered emotions that are recognisable from concrete situations in daily life. This creates awareness for the participants' development of consciousness regarding their search for meaning and the impact thereof on their daily lives. Action insight!

'I felt how much River struggles with the fact that it cannot pass through Sand. When I'm busy with something then I want to be able to continue, when something stops me, or when something intervenes, it gets on my nerves'. Another participant joins the conversation and says: 'I have that too, I played a River that said: 'I'm doing well right now, I feel the drive flowing in me and now I have to stop'. Then I, too, get angry at Sand who lies in the way.'

'As 'Sand' I really felt too much, like I wasn't allowed to be there. Even when I listened to those other Rivers, I felt confirmed in my identity of obstructor, something I recognise from daily life'. The director intervenes for a moment, correcting: 'You weren't assigned the identity of obstructor, that's what Sand felt when River came rolling in . . . : 'labeled as an obstructor'. Do you ever have that feeling?' The participant replies: 'No, I don't feel like an obstructor, but if you play Sand you do feel like that. Why is Sand not allowed to be there? As human beings, we all have the right to be respected, don't we?'

A very different kind of experience is expressed by another participant who played the role of Sand. 'I liked being sand. I thought it was great to stop the river that just kept on rolling'. The director asks: what did you find so 'great' about it? The participant replies: 'The feeling of strength, of personal strength. In daily life, I don't always dare to stop someone when I think that person has crossed a line. When that happens, I keep quiet, I let it pass . . . In cases like those, I would like to be like Sand. It was good to feel what power I had.'

## 4. Conclusions, Discussions and Recommendations

Now that we have presented the theoretical background of bibliodrama in Part I, and provided examples of real-life bibliodrama sessions in Part II, we will connect theory and practice in this concluding part. We are well aware that no general conclusions can be drawn on the basis of the few examples we have given. Our recommendations will focus on the possibilities of bibliodrama practice in an educational setting.

### 4.1. Conclusions

Based on our definition of bibliodrama and the experiences in the group of 18–20-year-old students of the Odisee Hogeschool (Odisee University of Applied Sciences) of Brussels (Belgium).we conclude that dramatisation, role-play and the imaginative practice of 'as if' are key ingredients in a group process of identity development that takes place in bibliodrama sessions through the interpretation and performance of stories from a narrative tradition. The method of bibliodrama is characterised by a preparation phase followed by three phases of the staged drama: warming up, acting and sharing. By 'staging' and acting, insight is gained into the nature of the characters that are being role-played, and, as a consequence, the participants may gain insight in their own positionality vis-à-vis the theme that is at stake in the bibliodrama. The facilitating remarks and comments of the leader of the group (the facilitator/'director') together with the sometimes unexpected and surprising interventions of participants function as disruptive moments [19,20], inviting a person to leave her/his comfort zone and take another until then unknown perspective on the situation and on her/himself. By consequence students become aware of the multifacetedness and multivoicedness of their self and their positioning process regarding world view traditions. This latter aspect is verbalised in the so-called sharing phase. After shaking off their roles, the participants share first the feelings they experienced in their role and, secondly, the insights that came to the fore and what the performance and the gained new insights mean to them as a person in their daily practices. No evaluative comments, no judgments are expressed in this phase. The focus is exclusively on sharing and listening. This is the way bibliodrama casts its influence in everyday life.

### 4.2. Discussions

For students, bibliodrama offers interesting opportunities to become aware of tense situations in their daily lives and to become aware of the fact that contemporary dilemmas, examined from a meta-perspective, resemble dilemmas described in age-old narratives—whether they are part of religious or secular tradition(s).

This article describes a single example of bibliodrama practice, an exploration of a single scene from a narrative. This one example shows the strength of bibliodrama, which is not just about

role-playing, but also about role-taking and acting based on a person's own intuitive and associative interpretations of the characters in a narrative.

In the examples we saw that the participants—as actors—can experience emotional inclinations towards actions they never considered before. Like River saying to Sand: 'You don't know me yet!' or Sand admitting: 'That's right River, I don't know you, but I know me. The way you come pouring from the mountains, that's something worth seeing, I don't want you to disappear. We must deliberate.'

*4.3. Recommendations*

Bibliodrama can be enriched by giving the audience a more active role, i.e., more active than just observing. The audience might be invited to 'double' a character on stage, by whispering inner thoughts they expect the character to have [12] (p. 30 ff). When River says to Sand 'you don't know me yet!', for example, a 'double' by a member of the audience might be 'and I don't want to be known by you, Sand!' A 'double' should always be verified by the 'director' with the question 'Is this in line with what you think or feel?' A 'double', whether or not approved, will contribute to the growing insight of the actor.

Another way to enrich a bibliodrama is by 'mirroring' the scene to one of the actors on stage. In such cases, a member of the audience is invited to copy the actor's position exactly and the actor is asked to watch that mirrored performance. What does she/he see and what kind of feelings does the scene evoke? Looking in the mirror, being confronted with one's own verbal and nonverbal expressions can be a disruptive experience, stimulating awareness and creating insight at the same time.

A third way to extend and deepen the process of dramatisation is through role change. This means that River takes on the role of Sand, and vice versa. To see how the opposite actor takes on the role (role-taking) you have played (role-playing), can result in a moment of wonder or recognition [12] (p. 35).

These three ways of enriching and deepening a bibliodrama are strongly reminiscent of psychoeducation and therapeutic applications of bibliodrama. The kind of insights that emerge during bibliodrama offer interesting starting points for psychoeducation about identity development and the multivoiced character of identity as described in Dialogical Theory. In addition, Dialogical Self Theory provides a challenging method—the so-called Self Confrontation Method—to explore the emotional relationships between the variety of 'voices' in the multivoiced society of mind in a verbal way (see [2,7]).

To make a bibliodrama a success, the expertise of the session leader is pivotal. She/he is the 'director' of the play, to whom a variety of essential tasks are entrusted: asking sophisticated questions to stimulate the imagination before the performance, giving finely tuned directions for role-play, being sensitive to dramatic experiences of participants that need a more therapeutic approach and keeping an eye on the group dynamics [12] (p. 115).

Bibliodrama, in our opinion, creates a safe space for the exploration of, and communication with, a plurality of narratives from a plurality of life orientations, like narratives from the Qur'an, the Bible, the Bhagavad Gita or the Anansi stories. Learning about the dogmatic content of traditions is only one aspect of interreligious teaching and learning, and probably a less inspiring aspect at that. Experiencing the actual meaning of these traditions requires a different approach, and in our view the method of bibliodrama offers interesting practical methods for experiential learning from traditional narratives. Students' own experiences with role-playing and role-taking will inspire and motivate them in daily life to practice their action insights. Bibliodrama, as shown above, enables young people to learn with and from each other [21]. Their creative, playful communication in their peer group with the story and with each other makes them smarter, not only cognitively but also—and especially—in an emotional and empathic way.

Bibliodrama as an open and safe space in a plural school community contributes to intercultural and interreligious understanding. The exploration of narratives and characters from 'other' cultural and religious traditions make them a part of the explorer's own multivoiced self. Integrating 'the other'

into one's own 'society of mind' is the basis for mutual respect in a context characterised by diversity in our secularised/secularising world.

**Funding:** This research received no external funding.

**Acknowledgments:** Sincere thanks to Stijn Van Tongerloo for the critical proofreading of this text, and for supplying improvements (both textual and content-wise) to make this article an integrated whole.

**Conflicts of Interest:** The author declares no conflict of interest.

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
