# Peer review of "Bibliodrama: Introducing Stories from Narrative Traditions in the Development of Young People’s Life Orientation"

_education, doi:10.3390/educsci9020107_

Round 1

Reviewer 1 Report

A fine article.  It is well written and connects a teaching and learning approach to a real problem in adolescent learning.  Well done.  My only suggestion is at the end of the paper.  I have seen and used bibliodramas many time in teaching.  I think there is a need for a preparation stage.  After participants have heard the story, they need time to prepare before they engage in the bibliodrama.  For me, there are times when this preparation must also include historical/ critical exegesis.  I wish you would mention the preparation stage and how simple it can be and also how complex.  

Author Response

Respons to Reviewer 1:

Thank you for your constructive comment. The author has added the preparation stage.

Reviewer 2 Report

 The paper was far too descriptive and contained a number of sweeping statements without supporting evidence. At times the author/s were informal and noted 'asides' which is inappropriate  for an academic paper. Too much of the paper was concentrated on explaining bibliodrama. The style of writing was convoluted and the paper was sadly quite uninteresting.

Abstract

Please remove the footnotes completely from the abstract. These should not appear nor should references.

Introduction

The introduction was not especially engaging and  the first section appeared to give too much explanation on the one theme, which made it rather boring.l

Please drastically reduce the number of footnotes as it is distracting for the reader. If you feel there is a need for further explanation then add these into the text. For example in line 68 , it is not necessary to refer to another author as it does not add anything to the text.  This occurs again in line 294. This style of informality is not appropriate for an academic paper. The introduction needs to be extensively revised and contain more themes.

Bibliodrama: Theory and Method

Please remove the quote from Shakespeare at the beginning of this section as it appears slightly pretentious. The tone of this section at times  is too informal, for example, the author/s give personal opinions on bibliodrama in footnote 7 which is more of an aside. If the author/s believe that this was important this should be included in the main text and supported by  a discussion and appropriate references.

Line 201 is in italics with no reason. There is also a lack of references around the rationale for using bibliodrama as a method. This section needs to be substantially rewritten.The use of  profanity does not add anything to the text. It is rather insulting to assume that young people have  limited vocabulary to apply when describing emotions. There was no reference in the text to any studies to support this opinion. The second section over-explained bibliodrama and just seemed to ramble at times. The inclusion of Winnicott,(1971)'s work as an example of bibliodrama was misplaced and did not seem to align with bibliodrama.  The rest of this section until section 3 seem to just repeat in a different ways bibliodrama in practice and could be reduced substantially.

p.p1 {margin: 0.0px 0.0px 0.0px 0.0px; font: 10.0px Helvetica}

Bibliodrama in Action – Interreligious Identity Construction

It was not till line 667 it was explained to the reader who the participants were.  The story was outlined earlier but it was quite confusing as there seemed to additional explanations given which were not particularly interesting.

There was no rationale as to why the facilitator had to be an expert in hermeneutics, indeed earlier the author /s stated that following a religion was not necessarily essential. 

The verb tenses were inconsistent and it made this section difficult to follow. ~It would have been helpful if participants' comments were written in italics .

The outcomes , discussion and recommendations

This was rather brief and did not seem to come to any conclusion  except that bibliodrama was useful for students to practice as it was a safe place. But as this article only gave one example in practice it is difficult to see why an educational authority would decide to adopt it.

Author Response

Respons to Reviewer 2:

Thank you for your constructive comment. The author has removed most of the footnotes  and included their meaning in the tekst.

The quote of Shakespeare is by the author seen as adding to the text. However, the quote is situated at an other part in the text.

The italics, meant to emphasize that part of the text, are removed.

At an earlier stage in the text the characteristics of the research population are mentioned.

The need for expertise of the facilitator in hermeneutics is clarified.

The outcomes are described in a more nuanced way, being aware of the impossibility for generalisation of a case study.

Reviewer 3 Report

The topic of the 21-page study fits well into the contemporary scientific profile.  However, the problem is that the references to the conclusions and results are not explicit enough.

The second part of the manuscript does not contain any new results beyond the descriptive description of the descriptor, the method and its use. Please clarify the reference to the narrative in line 194: "...narratives (for example: biblical, Qur'anic, ...)".

The final chapter, where the readers can find the conclusions and suggestions. These were defined as tighter, even in the short term. In particular, the conclusions contain generalities.

Due to the curiosity and methodological importance of the topic, I recommend the revision of the study. The equalization of the structure can be solved by expanding the analytical evaluation parts and reducing the descriptive parts.

The more specific definition of results and conclusions is essential.

The list of references is a modest list of 19 sources, most of which have been published in Flemish. Finally, it is suggested to increase the number of references to the sources, especially the number of English-language publications.

The equalization of the structure can be solved by expanding the analytical evaluation parts and reducing the descriptive parts

Author Response

Respons to reviewer 3:

Thank you for your constructive comments.

The conclusion and recommendation part is improved by the author.

The analytical part is more specific, referring to quotes of students, and the descriptive part is shortened substantially.

Due the fact that the author is Flemish and not fluent in English, some of the references are in Flemish. 

Round 2

Reviewer 2 Report

please remove first names when citing authors . as in Hannah Arendt (2013) listening, exploring, questioning, accepting and respecting, copying, recognising, re-phrasing, distinguishing, conflicting and hyphenating. Please reference and explain how this is relevant to your article as your readers may not be familiar to these terms when applied to a young person's development (see also Verhofstadt-Denève, 2003)Could you stop putting in the phrase 'see' also' Just put in citation. Although you have reduced the number of footnotes , please reduce the explanation in footnote1. If you feel it is important to justify your opinion then this should be in the main text. And again please stop suggesting to the reader who they should read.

Correct application of brackets of the following  (DST; Hermans & Hermans-Konopka, 2010) provides. Sp  dilemma’s . Add in page number ‘Is this not a secondary citation? thepersonality dynamic or the potential, that needs to be nurtured and developed in a person; Is this not a quote?and is it not from Winnicot ?  Can you avoid using ibid as it is not clear which reference you are referring to.

I appreciate that you have reduced the word count, but section 2 could be drastically reduced as it is currently over 8 pages. 

There is no critical analysis of bibliodrama. You do not address how this could affect young people who  may be experiencing issues in their own life . Although you state that their reactions are not  to be discussed after the event ,  it would have been helpful if you had provided evidence of the longer term effects of this technique on the participants. 

Author Response

Dear Reviewer,

Thank you for your attentive reading of the second draft of the text. 

In the references 'see also' is removed in some cases. However, in some cases according to the author it is relevant to refer 'see also', because the author referred to does not write about the aspect at stake in exactly the same way, but in a way that sheds an interesting light on that aspect.

The first name of Hannah Arendt is used when she is referred for the first time. All other times only the family name is mentioned. The author prefers to use her first name the first time since the combination of her first and family name by many authors is perceived as one whole.

Regarding the verbs listening, exploring etc the author now refers to Marcia's stages of exploration and commitment.

Footnote nr. 3 is substantially shortened.

Regarding DST and Hermans&Hermans-Konopka: the reference relates to the whole publication on multi-voicedness and its dialogical character.

When 'ibid' is used, this refers to the author that is mentioned just before in that section of paragraph. The author checked for the adequateness of 'ibid's used.

Section 2 is shortened. The author is of the opinion that shortening this section even more would make it difficult for a layperson to follow the line of thought and the practice of the manyfold aspects of bibliodrama.

The author agrees with the reviewer that it would be very interesting to know more about the longterm effects of bibliodrama. However, a follow up of the participants of the bibliodrama was not included in this project.

Reviewer 3 Report

The re-edited paper is well structured, and the theoretical overview is becoming much more pointed. Description of the empirical part of authors’ work more clear, represents an innovative methodological implementation. To add a few new English references into the list of literature giving help for the further orientations to the readers and other researches in this field.
The Bibliodrama is very interesting and surely will attract all the student towards the development of students’ life orientation.

Author Response

Dear Reviewer, thank you for your comments in the 2nd round.

kind regards,